# The emergence of Omicron VOC and its rapid spread and persistence in the Western Amazon

**Gabriella Sgorlon**[1,2,3], **Tárcio P. Roca**[1,3,4], **Ana Maisa Passos-Silva**[1,2,3], **Jackson A. S. Queiroz**[1,2,3], **Karolaine S. Teixeira**[1,3], **Adrhyan Araújo**[1,3], **Flávia S. Batista**[5], **Valquiria R. Souza**[6], **Franciane M. Oliveira**[6], **Luis G. Morello**[7,8], **Fabricio K. Marchini**[7,8], **Juan M. V. Salcedo**[1,2,3], **Rita de Cassia P. Rampazzo**[7], **Felipe G. Naveca**[9], **Deusilene Vieira**[1,2,3]*

1 Laboratório de Virologia Molecular, Fundação Oswaldo Cruz Rondônia - FIOCRUZ/RO, Porto Velho, RO, Brazil, 2 Programa de Pós-Graduação em Biologia Experimental, Universidade Federal de Rondônia - UNIR, Porto Velho, RO, Brazil, 3 Centro de Pesquisa em Medicina Tropical, CEPEM, Porto Velho, RO, Brazil, 4 Laboratório de Hepatites Virais, Instituto Oswaldo Cruz/IOC, FIOCRUZ, Rio de Janeiro, RJ, Brazil, 5 Coordenação Estadual do COVID-19, AGEVISA/RO, Porto Velho, RO, Brazil, 6 Instituto de Medicina Tropical da Faculdade de Medicina da USP, São Paulo, SP, Brazil, 7 Instituto de Biologia Molecular do Paraná - IBMP, Curitiba, PR, Brazil, 8 Instituto Carlos Chagas (ICC), FIOCRUZ-PR, Curitiba, Paraná, Brazil, 9 Laboratório de Virologia do Instituto Leônidas e Maria Deane, FIOCRUZ/AM, Manaus, AM, Brazil

* deusilene.vieira@fiocruz.br

**Data Availability Statement:** All relevant data are within the manuscript and its Supporting Information files.

## Abstract

Genomic surveillance represents a strategy to understanding the evolutionary mechanisms, transmission, and infectivity of different SARS-CoV-2 variants. We evaluated 603 individuals positive for SARS-CoV-2 from 34 municipalities of Rondônia between December 2021 to December 2022. Nasopharyngeal samples were collected, RNA was extracted and screened using RT-qPCR for VOCs. RNA of the samples were sequenced and further analyzed for phylogeny, mutations, and lineages, totaling 96.19% of samples positive for Omicron VOC in this cohort. We observed that most individuals had at least two doses, however 18.97% were not vaccinated with any dose. 554 sequences were amenable to analysis for alignment and phylogenetic characterization; this group corresponded to the 27 subvariants of the Omicron VOC; a total of 100 mutations were identified, 48% of which were found in the S gene. In conclusion, the data demonstrated the rapid spread and persistence of Omicron VOC in Rondônia during the 12-month study period. Although high frequency of mutations was found in the analyzed samples, there were no individuals with a severe clinical profile, demonstrating that vaccination had a positive effect in those cases.

## Introduction

Severe Acute Respiratory Syndrome Virus 2 (SARS-CoV-2) has the RNA genome with high mutational rate [1, 2]. To date, about 12,000 mutations have been reported, some are related to increased infectivity, vaccine-escape and worsening of the clinical presentation. [1–3].

**Funding:** This study was funded by Fundação Oswaldo Cruz de Rondônia – FIOCRUZ/RO, Departamento de Ciência e Tecnologia (DECIT), Fundação para o Desenvolvimento da Ação Científica e Tecnológica e à Pesquisa do Estado de Rondônia - FAPERO (Process:350.095.442.048.526.000.000/ 2016; Public bid invitation: 012/2016 PRO-RONDÔNIA and 001/2020 PPSUS) and by Instituto Nacional de Epidemiologia da Amazônia Ocidental - INCT EpiAmO. FGN is a CNPq fellow. Departamento de Ciência e Tecnologia (DECIT) of the Brazilian MoH, US/CDC and OPAS, Brazilian office. The funders had no role in study design, data collection and analysis, decision to publish, or preparation of the manuscript.

**Competing interests:** At the time of submission, R. C.P.R, L.G.M and F.K.M were employees at IBMP, which manufactures and commercializes the test described in this study. The other authors declare no potential conflict of interest. This does not alter our adherence to PLOS ONE policies on sharing data and materials

Genomic surveillance is an appropriate approach that has supported the investigation and circulation of cases of new variants and subvariants in circulation to support government decision making and implementation of health measures aimed at controlling SARS-CoV-2 infections [3–6].

Omicron VOC identified in November 2021 in South Africa, has approximately a total of 32 mutations concentrated mainly in the receptor binding domain (RBD) and Spike protein (S). This VOC spread rapidly to 40 countries after only one month of its emergence [2, 7].

In Brazil, the first case of Omicron VOC was reported in November 2021 in the state of São Paulo, remaining the dominant VOC throughout 2022 with 13,986,090 confirmed cases and 77,173 deaths in the country. Genomic surveillance detected that by February, the Omicron variant was already dominant, reaching 99.8% of the samples analyzed [8–10].

In the first week of the year 2022 there was an increase in the number of cases of COVID-19, demonstrating an atypical profile in Northern Brazil. This fact could be related to the entry of Omicron VOC and its sublines with high transmissibility rates [11, 12]. Thus, the aim of this study was to evaluate the epidemiological and genomic profile for confirmation about the circulating SARS-CoV-2 variants in the Brazilian Western Amazon.

## Materials and methods

### Ethical aspects and study site

This study was conducted in the Laboratory of Molecular Virology at Fiocruz/RO, with approval from the Research Ethics Committee of the Research Center for Tropical Medicine of Rondônia-CEPEM/RO 4.000.086 and was carried out in accordance with the ethical principles stipulated by the 1975 World Medical Assembly and the Ministry of Health (Resolution 466).

All experiments were conducted in accordance with the relevant guidelines and regulations and were exempted from the requirements for informed consent due to the pandemic timing making direct contact with the subjects impossible. In addition, the study poses no risk to subjects due to the use of RNA from naso- or oropharyngeal samples already collected and extracted for the diagnosis of COVID by the centers, with no need for additional collections.

### Biological samples and epidemiological data

The cohort of 603 individuals positive for SARS-CoV-2 were selected by convenience from primary care clinics and reference centers in different municipalities of Rondônia state between December 2021 to December 2022. Diagnosis of SARS-CoV-2 was carried out in Laboratório Central de Saúde Pública de Rondônia (LACEN/RO) by RT-qPCR with One Step/ COVID-19 kits (IBMP, Brazil). Epidemiological data and vaccination status were collected from medical records in the GAL/RO, SIVEP-Gripe and E-SUS databases.

### Extraction of viral RNA

RNA was extracted from 140 µL of samples collected using nasopharyngeal swab in viral transport medium; using QIAamp® Viral RNA Mini Kits (QIAGEN, Germany) according to the manufacturer's instructions. RNA from each sample was eluted in 60 µL of AVE buffer for viral load and inference testing.

### Screening for Alpha, Beta, and Gamma VOCs

In order to screen for the Alpha, Beta and Gamma VOCs, the multiplex RT-qPCR protocol of Vogels et al. was utilized [13]. Three targets were included in this multiplex: N1, deletion Δ69/ 70 and deletion of SGF Δ3675–3677 in the ORF1a gene.

The cycling process used for the reaction was 55˚C for 10 minutes for reverse transcription, PCR activation at 95˚C for 1 minutes, 39 subsequent cycles of 10 seconds at 95˚C and 31 seconds at 60˚C. Samples with Ct <35 for the N1 target alone were characterized as the Alpha VOC, and samples with Ct <35 for the N1 target and Δ69/70 deletion were classified as Beta or Gamma.

## Screening for the Delta VOC

The inference test was performed using the primers and probes (Table 1) described by Yaniv et al. with modifications [14]. The final reaction volume was 20 μL with a primer concentration of 0.5 μM and a probe concentration of 0.2 μM. The reaction contained 5 μL of RNA sample and the reaction steps were performed according to the manufacturer recommendations using TaqMan Fast Virus 1-Step Master Mix (Applied Biosystems 1, California, USA).

The cycling process used for the reaction was 51˚C for 10 min for reverse transcription, PCR activation at 95˚C for 1 min, 40 subsequent cycles of 10 s at 95˚C and 31 s at 60˚C; the final step included fluorescence capture.

## Screening for the Omicron VOC

All samples that were negative for Alpha, Beta, Gamma and Delta variants were subjected to RT-qPCR genotyping for the Omicron variant. The reaction was performed using 5 μL of Taq-Path™ 1-Step RTqPCR Master Mix (4x), 0.5 μL of TaqMan SARS-CoV-2 Mutation Panel Assay (40X) for the SNP assay S:K417N, 5 μL of extracted RNA and a final volume of 20 μL. The cycling used for the reaction was as follows: pre-reading at 60˚C for 30 seconds, Reverse Transcription at 50˚C for 10 min, DNA polymerase activation at 95˚C for 2 minutes, 45 cycles at 95˚C for 3 seconds for denaturation and 60˚C for 30 seconds for annealing and extension, ending with a post-reading at 60˚C for 30 seconds. The genotyping module of the Design & Analysis Software Version: 2.6.0 (Thermo Fisher Scientific) was used for analysis, with a 95% confidence interval for real-time data. Given that this mutation is described as prevalent only in subvariants of Beta, Delta and Omicron, all samples positive for S:K417N and negative for the other variants tested were classified as Omicron [15–17].

## Quantification of viral load

The viral load of samples was determined using 5μL of viral RNA extracted using the Multiplex One-Step RT-qPCR assay for detection of SARS-CoV-2 as developed by Queiroz et al, 2021 [18].

## Complete genome sequencing of SARS-CoV-2

The sequencing of the complete SARS-CoV-2 genome was performed with the support of the FIOCRUZ Genomic Surveillance Network. Samples with Ct values <25, based on quantitative assays, were selected to allow for high genomic coverage. Nucleotide sequencing was

**Table 1. RT-qPCR primers and probe for the Delta VOC.**

| Name | Description | Sequence 5'-3' |
| --- | --- | --- |
| Delta_CoV | Sense | GTTTATTACCACAAAAACAACAAAAG |
| Delta_CoV | Antisense | GGCTGAGAGACATATTCAAAAGTG |
| Delta_CoV | Probe | Cy3- TGGATGGAAAGTGGAGTTTATTCTAGT- BHQ 2 |

Adapted from: Karin Yaniv. Accessed on: July 5th, 2021.

performed using Illumina MiSeq or NextSeq platforms and the COVIDSEQ Kit (Illumina, San Diego, CA, USA) [19].

FASTQ reads were generated by the Illumina pipeline in BaseSpace. Consensus sequences were generated using DRAGEN COVID LINEAGE 3.5.1 to 3.5.3, according to the most updated version of this application in each sequencing run. Subsequently, the quality of the consensus files was analyzed using the Nextclade v1.5.2 tool [20], those with more than 1% "Ns" ambiguities had the FASTQ files imported into Geneious Prime 2021 for trimming and assembly using a custom workflow employing the BBDuk and BBMap tools (v38.84) and the NC_045512.2 RefSeq as a template with careful visual inspection. Using both approaches, we generated consensus sequences with average depth coverage greater than 800X, excluding duplicate reads. The genome-wide consensus sequences of SARS-CoV-2 were initially assigned to viral lineages using the Pango Lineage web application [21].

### Data acquisition and Maximum-Likelihood (ML) phylogeny

Available high quality (>29 kb) whole genomes (<1% of N) of BA.* sampled in Brazil (n = 363) were downloaded from the GISAID EpiCoV database on January 5th, 2023. The sequences were aligned using MAFFT v.7.487 [22]. The best model of nucleotide substitution was measured (GTR+G+I) using ModelFinder [23] and the phylogenetic tree was reconstructed using the maximum likelihood method in the program IQ-TREE v.2.1.3 [24]. Branch support values were obtained using Ultrafast Bootstrap with 1,000 replicates. The tree was visualized and edited with FigTree v.1.4.4 [25]. SARS-CoV-2 genomes were classified into lineages using the available software Pangolin [26] and mutations were analyzed with Nextclade Beta [27].

### Statistical analysis

Descriptive analyses were represented through central tendency and dispersion measurements. A Chi-square test was used for statistical inference with a significance level of 5% ($p < 0.05$). Statistical analysis was performed and graphics were generated using the software R v4.0.3.

### Results

In the cohort of 603 SARS-CoV-2 positive samples, 96.19% (580/603) corresponded to the Omicron VOC and 3.81% (23/603) to the Delta variant, without detection of either Gamma, Alpha or Beta VOCs; the data were confirmed by three RT-qPCR screening assays for VOCs described in methodology.

These cases were identified since December 1, 2021, to December 7, 2022 in 34 municipalities in the state of Rondônia (65.38%), demonstrating a wide distribution of the variant. The first collection period, December 2021 and January 2022, we observed higher prevalence in the North, West and South regions of state; sequentially, in the second period, February 2022 and March 2022, notifications came from the most populous municipalities in the state, such as Porto Velho, Ji-paraná and Ariquemes.

After the periods described on the map, we continued genomic surveillance until December 2022 and identified another 167 sequences characterized as Omicron, in which 58.68% are predominantly from the state capital,, with no other variants identified.

The median age of the population analyzed was 39 years old (SD 16.74), with ages ranging from 5 month to 95 years old; 50.69% (294/580) were female and 49.31% (286/580) were male. The main symptoms reported by patients included headache in 49.83% (289/580), cough with 49.48% (287/580) and fever in 42.24% (245/580) of cases. Dyspnea, disturbance of smell and

taste were less reported symptoms with 7.24% (42/580), 2.59% (15/580) and 2.41% (14/580) of patients, respectively. Only 5.86% (34/580) of patients fell into the asymptomatic group.

Regarding vaccination, 18.97% (110/580) were not vaccinated, and among those vaccinated, 7.76% (45/580) had only one dose, 43.28% (251/580) with two, 26.21% (152/580) with three, and 3.79% (22/580) with four doses. The immunizers received by the study population were COVISHIELD, Sinovac, Comirnaty® | Pfizer Brazil, and Janssen-Cilag. There were 5 deaths in this study, 3 unimmunized individuals with a mean age of 75 years and 2 with the first and second dose of vaccine with a mean age of 93 years. The deaths were from individuals infected by subvariants BA.1 (1/5), BA.1.1 (3/5), and BA.5.1 (1/5). Individuals had a mean of 5 days (SD 3.6 days) of symptoms until diagnosis. The viral load of the cohort had an interquartile median of 7.08 Log10/mL, and less than 0.52% (3/580) of individuals had a quantifiable viral load after 14 days of symptoms and one individual symptomatic up to 19 days (Fig 1).

Among the 603 samples tested by RT-qPCR, 580 samples were selected for sequencing, taking into consideration the viral load and multiplex tests for exclusion of other VOCS. Of the sequenced samples 95.52% (n = 554) had good quality metrics for phylogeny and mutation analysis. Fig 2 demonstrated the maximum likelihood phylogeny in relation to the clade classification of the Omicron variant.

The analyses of the sequenced samples corresponded to 30.14% (167) BA.1; 18.05% (100) BA.1.1; 6.14% (34) BA.1.1.1; 0.18% (1) BA.1.1.14; 0.18% (1) BA.1.1.15; 0.18% (1) BA.1.1.18; 1.26% (7) BA.1.14; 12.82% (71) BA.1.14.1; 0.18% (1) BA.1.14.2; 0.90% (5) BA.1.15; 2.17% (12) BA.1.17.2; 0.18% (1) BA.1.20; 0.36% (2) BA.1.9; 2.35% (13) BA.2; 0.18% (1) BA.2.12.1; 0.18% (1) BA.2.36; 0.72% (4) BA.2.56; 0.18% (1) BA.2.81; 1.81% (10) BA.4; 0.18% (1) BA.4.1; 9.93% (55) BA.5.1; 0.18% (1) BA.5.1.15; 8.84% (49) BA.5.2.1; 0.36% (2) BA.5. 6; 0.54% (3) BE.9;

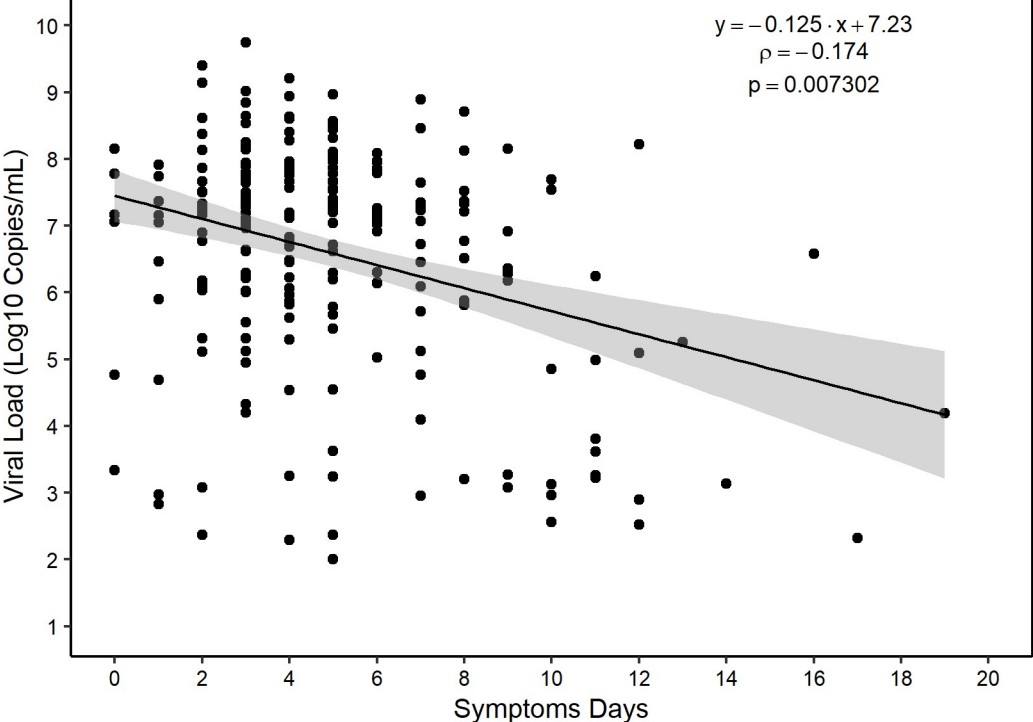

$$y = -0.125 \cdot x + 7.23$$
$$\rho = -0.174$$
$$p = 0.007302$$

**Fig 1. Viral load measured by the number of days after onset of symptoms.**

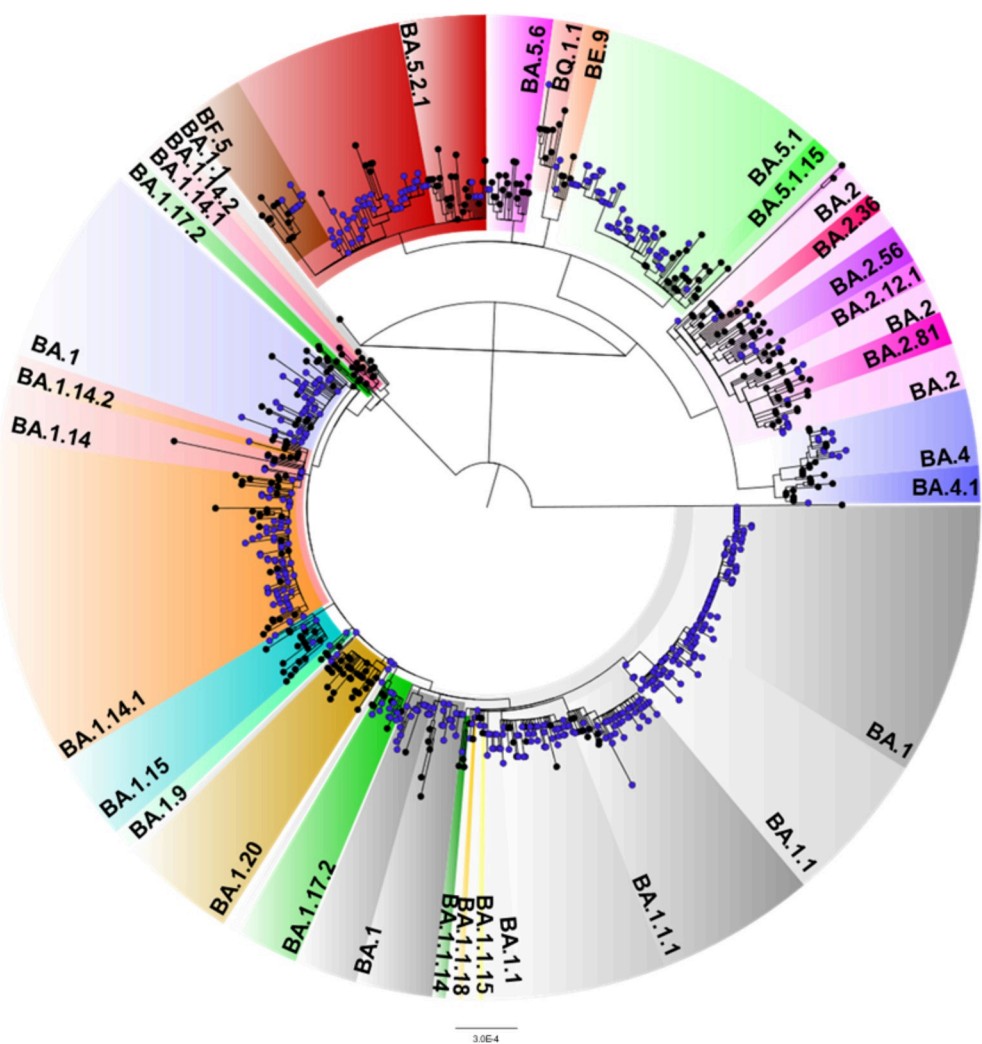

**Fig 2. Maximum likelihood phylogenetic tree presenting 554 sequences obtained in this study and 363 genomes retrieved from GISAID.** The samples are indicated as blue circles. The tree was rooted with the most ancestral sequence (EPI_ISL_402123).

1.62% (9) BF.5 and 0.18% (1) BQ.1.1. To date, no cases of XBB.1.5 have been reported in the state of Rondônia.

We identified 100 mutations (Fig 3) with a frequency greater than 2%. The S gene presented 48% (48/100) mutations, followed by 24% (24/100) in ORF1a, 8% (8/100) in ORF1b, 7% (7/100) in N, and 13% (13/100) distributed among the other genes (ORF3a; M; E; ORF6 and ORF9b).

Mutations previously identified as signatures for the Omicron VOC genomes were analyzed. The N gene region presented four mutations (*E31-32del*, *G204R*, *R203K* and *P13L*) and the ORF1a gene three mutations (*T3255I*, *P3395H* and *S3675-3677del*) with a frequency of 100%. The *N679K* and *P681H* substitutions, considered signature over other variants, showed a frequency of 100%, along with a substitution in *N501Y* present in 99.64% of the sequences, all of which were characterized as changes in the Spike protein. Six other mutations (*D614G*, *D796Y*, *G339D*, *H655Y*, *N764K*, *N969K*, and *Q954H*) were found in this region at high frequency.

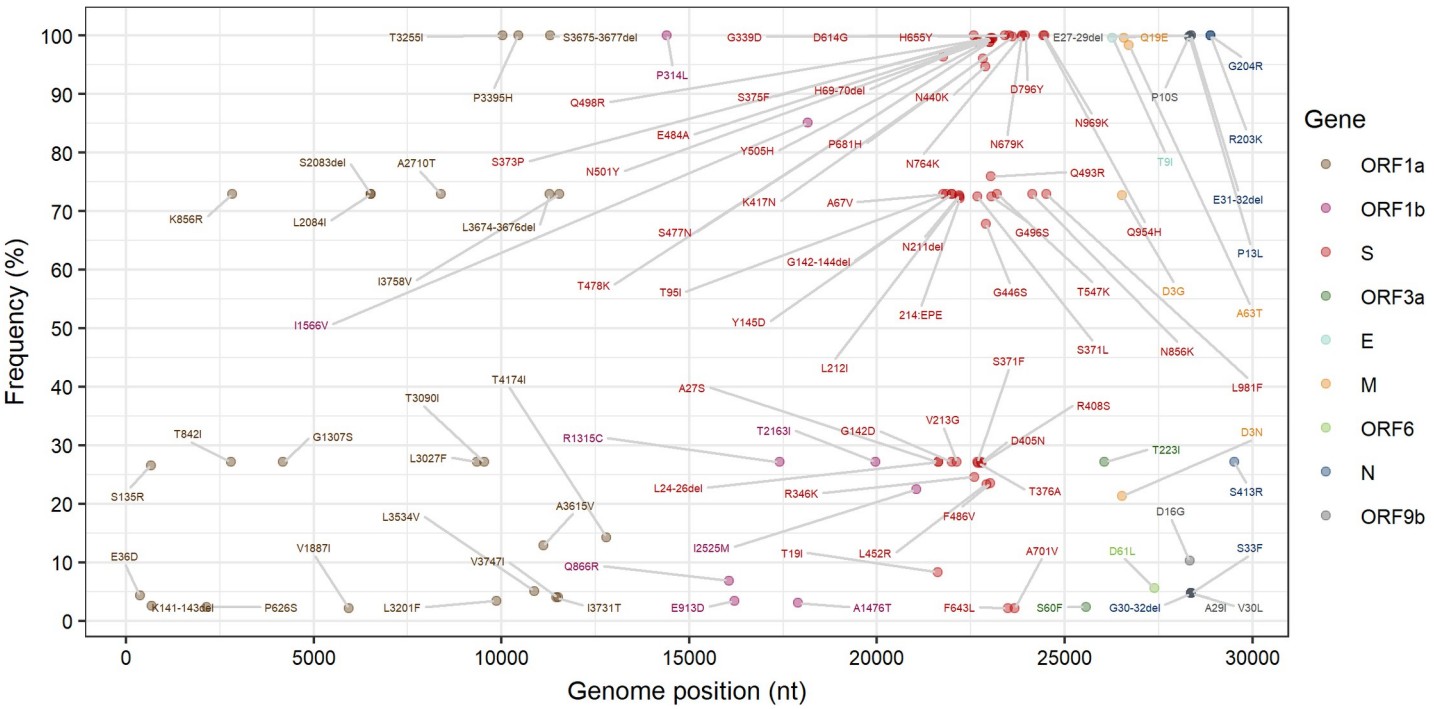

**Fig 3. Dot plot demonstrating the frequency of mutations identified in the set of 554 samples analyzed by NGS.** The labels of mutations with frequencies <2% have been hidden.

## Discussion

The emergence and persistence of Omicron VOC worldwide was also evaluated in the state of Rondônia, located in the Northern Region of Brazil. The first case in the state was identified on December 1, 2021, and then we continued the analysis until December 2022, completing a total of 580 samples consistently characterized through genomic surveillance.

In parallel with the emergence of Omicron VOC, questions have arisen regarding the frequency of symptoms reported during the period of infection. In this study, the most described symptoms were similar to those detected in individuals infected by other variants, as infected presenting headache, cough and fever. Unlike the first SARS-CoV-2 cases reported in the early 2020s, a low frequency of individuals infected with Omicron VOC demonstrated symptoms such as taste and smell disturbances [28, 29].

In this cohort, 81% of the individuals received at least 1 dose of the COVID-19 vaccine and there is evidence that immunization reduced the more severe cases. However, it is worth noting that the 5 deaths recorded were from unvaccinated individuals or without booster doses, presenting advanced age, highlighting the importance of following vaccination protocols [30].

Individuals infected with the Omicron variant had high viral load up to 9 days after symptom onset, with values of this parameter decreasing from 10 days [31, 32]. Similarly, the cohort in this study showed high levels and maintenance of viral load for up to 10 days after symptom onset, with a mean of 5 days.

The Omicron variant was first detected in the state of Rondônia in December 2021 while the Delta variant was still predominant. After the introduction of Omicron, it subsequently became predominant, accounting for approximately 100% of the samples analyzed in February 2022. A previously published review article demonstrated that Omicron average basic and effective reproduction numbers were 8.2 and 3.6, respectively, meaning a 2.5- to 3.8-fold

higher transmissibility than the Delta variant, which may be partially explained by the higher number of mutations, higher transmissibility and greater capability for immune escape [33].

Even with the largest portion of the study population with full immunization, Our findings showed the presence of the 27 subvariants of the Omicron VOC, evidencing that the profile of this VOC exhibits reduced susceptibility to vaccine-induced neutralizing antibodies and risk of new waves of infection [34]. One study showed that neutralizing activity was reduced against BA.2.12.1 and BA.4/BA.5 compared to BA.1 [35]. Another study showed that the BQ.1.1 subvariant showed enhanced resistance against sera from healthcare workers vaccinated with 3 doses [36]. Evidence demonstrating the need for protection against future SARS-CoV-2 variants.

The Omicron variant has a highly transmissible profile due to a higher number of mutations than the other variants [37]. Mutations in the Spike protein play a signature role in other VOCs, such as *N679K*, *P681H*, and *N501Y* were visualized in all sequences, and this persistence of mutations is directly linked to the increased rates of infectivity, the high transmission capacity, and the potential for rapid dispersal of this variant [38–41].

This characteristic is visualized among popular gene variations, such as *D614G*, which was found with high frequency among the sequences, reported in studies that analyze the epistatic interaction aiming at the transmissibility that the virus achieves in infection [42–44].

The Omicron sequences analyzed here carried the *H655Y* mutation in a high proportion when compared to other alterations, this same mutation was also characterized in variant B.1.1.33 [45], characterizing with a common mutation between the strains.

In conclusion, the data demonstrated the rapid spread of Omicron VOC by the state of Rondônia, Western Amazon region, throughout 2022 in a dominant manner, with 27 subvariants. Although there were a high number of mutations in the sequenced samples, the patients did not present a severe clinical profile, demonstrating that vaccination had a positive effect in these cases. In addition, there was no identification of XBB.1.5.

## Supporting information

**S1 Table. SARS-CoV-2 genomes.** All genome sequences and associated metadata in this dataset are published in GISAID's EpiCoV database. To view the contributors of each individual sequence with details such as accession number, Virus name, Collection date, Originating Lab and Submitting Lab and the list of Authors, visit 10.55876/gis8.230107pn.
(PDF)

## Acknowledgments

The present study was developed by a group of researchers from Laboratório de Virologia Molecular da Fundação Oswaldo Cruz, in collaboration from Coordenação de Aperfeiçoamento Pessoal de Nível Superior–CAPES, from whom some authors received financial aid (scholarships) during the production of this study, the Vice president of Vigilância em Saúde e Laboratórios de Referências of Fiocruz, Instituto de Biologia Molecular do Paraná (IBMP) and Laboratório Central de Saúde Pública de Rondônia (LACEN/RO) were essential for the development of the study.

## Author Contributions

**Conceptualization:** Gabriella Sgorlon, Rita de Cassia P. Rampazzo, Deusilene Vieira.

**Data curation:** Gabriella Sgorlon, Tárcio P. Roca, Karolaine S. Teixeira.

**Formal analysis:** Gabriella Sgorlon, Tárcio P. Roca, Rita de Cassia P. Rampazzo, Deusilene Vieira.

**Funding acquisition:** Luis G. Morello, Fabricio K. Marchini, Juan M.V. Salcedo, Felipe G. Naveca, Deusilene Vieira.

**Investigation:** Gabriella Sgorlon, Rita de Cassia P. Rampazzo.

**Methodology:** Gabriella Sgorlon, Tárcio P. Roca, Jackson A. S. Queiroz, Felipe G. Naveca, Deusilene Vieira.

**Project administration:** Deusilene Vieira.

**Resources:** Deusilene Vieira.

**Supervision:** Rita de Cassia P. Rampazzo, Deusilene Vieira.

**Writing – original draft:** Gabriella Sgorlon, Tárcio P. Roca, Ana Maisa Passos-Silva, Jackson A. S. Queiroz, Karolaine S. Teixeira, Adrhyan Araújo, Deusilene Vieira.

**Writing – review & editing:** Flávia S. Batista, Valquiria R. Souza, Franciane M. Oliveira, Luis G. Morello, Fabricio K. Marchini, Juan M.V. Salcedo, Rita de Cassia P. Rampazzo, Felipe G. Naveca, Deusilene Vieira.

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
