## [Decision Letter · Decision Letter 0]

13 Mar 2023

PONE-D-23-01948The emergence of Omicron VOC and its rapid spread and persistence in the Western AmazonPLOS ONE

Dear Dr. Vieira,

Thank you for submitting your manuscript to PLOS ONE. After careful consideration, we feel that it has merit but does not fully meet PLOS ONE’s publication criteria as it currently stands. Therefore, we invite you to submit a revised version of the manuscript that addresses the points raised during the review process.

We look forward to receiving your revised manuscript.

Kind regards,

Babatunde Olanrewaju Motayo, Ph.D.

Academic Editor

PLOS ONE

Journal Requirements:

  "This study was funded by Fundação Oswaldo Cruz de Rondônia – FIOCRUZ/RO, Departamento de Ciência e Tecnologia (DECIT), Fundação para o Desenvolvimento da Ação Científica e Tecnológica e à Pesquisa do Estado de Rondônia - FAPERO (Process:350.095.442.048.526.000.000/ 2016; Public bid invitation: 012/2016 PRO-RONDÔNIA and 001/2020 PPSUS) and by Instituto Nacional de Epidemiologia da Amazônia Ocidental - INCT EpiAmO. FGN is a CNPq fellow. Departamento de Ciência e Tecnologia (DECIT) of the Brazilian MoH, US/CDC and OPAS, Brazilian office."

   "The present study was developed by a group of researchers from Laboratório de Virologia Molecular da Fundação Oswaldo Cruz, in Rondônia, with financial support from the Genomic Coronavirus Fiocruz Network, Departamento de Ciencia e Tecnologia (DECIT), Fundação para o Desenvolvimento das Ações Científicas 

e Tecnológicas da Pesquisa do Estado de Rondônia – FAPERO, Programa de Pesquisa para o SUS (PPSUS), as well as Instituto Nacional de Ciência e Tecnologia de 

Epidemiologia da Amazônia Ocidental – INCT- EpiAmo who have been important contributors to scientific development in the Amazon region. Collaboration from 

Coordenação de Aperfeiçoamento Pessoal de Nível Superior – CAPES, from whom some authors received financial aid (scholarships) during the production of this study, the Vice president of Vigilância em Saúde e Laboratórios de Referências of Fiocruz, Instituto de Biologia Molecular do Paraná (IBMP) and Laboratório Central de Saúde Pública de Rondônia (LACEN/RO) were essential for the development of the study"

   "This study was funded by Fundação Oswaldo Cruz de Rondônia – FIOCRUZ/RO, Departamento de Ciência e Tecnologia (DECIT), Fundação para o Desenvolvimento da Ação Científica e Tecnológica e à Pesquisa do Estado de Rondônia - FAPERO (Process:350.095.442.048.526.000.000/ 2016; Public bid invitation: 012/2016 PRO-RONDÔNIA and 001/2020 PPSUS) and by Instituto Nacional de Epidemiologia da Amazônia Ocidental - INCT EpiAmO. FGN is a CNPq fellow. Departamento de Ciência e Tecnologia (DECIT) of the Brazilian MoH, US/CDC and OPAS, Brazilian office."

   "At the time of submission, R.C.P.R, L.G.M and F.K.M were employees at IBMP, which manufactures and commercializes the test described in this study. The other authors declare no potential conflict of interest."

7. We note that you have included the phrase “data not shown” in your manuscript. Unfortunately, this does not meet our data sharing requirements. PLOS does not permit references to inaccessible data. We require that authors provide all relevant data within the paper, Supporting Information files, or in an acceptable, public repository. Please add a citation to support this phrase or upload the data that corresponds with these findings to a stable repository (such as Figshare or Dryad) and provide and URLs, DOIs, or accession numbers that may be used to access these data. Or, if the data are not a core part of the research being presented in your study, we ask that you remove the phrase that refers to these data.

8. Your ethics statement should only appear in the Methods section of your manuscript. If your ethics statement is written in any section besides the Methods, please delete it from any other section. 

9. We note that Figure 1 in your submission contain [map/satellite] images which may be copyrighted. All PLOS content is published under the Creative Commons Attribution License (CC BY 4.0), which means that the manuscript, images, and Supporting Information files will be freely available online, and any third party is permitted to access, download, copy, distribute, and use these materials in any way, even commercially, with proper attribution. For these reasons, we cannot publish previously copyrighted maps or satellite images created using proprietary data, such as Google software (Google Maps, Street View, and Earth). For more information, see our copyright guidelines: http://journals.plos.org/plosone/s/licenses-and-copyright.

10. Please include captions for your Supporting Information files at the end of your manuscript, and update any in-text citations to match accordingly. Please see our Supporting Information guidelines for more information: http://journals.plos.org/plosone/s/supporting-information. 

Additional Editor Comments:

You are advised to revise your manuscript in line with the reviewers comments. You are also to provide a step by step response to all reviewers concerns to the manuscript.

Reviewers' comments:

Reviewer's Responses to Questions

**Comments to the Author**

1. Is the manuscript technically sound, and do the data support the conclusions?

Reviewer #1: Yes

Reviewer #2: Yes

2. Has the statistical analysis been performed appropriately and rigorously? 

Reviewer #1: Yes

Reviewer #2: Yes

3. Have the authors made all data underlying the findings in their manuscript fully available?

Reviewer #1: Yes

Reviewer #2: Yes

4. Is the manuscript presented in an intelligible fashion and written in standard English?

Reviewer #1: Yes

Reviewer #2: Yes

5. Review Comments to the Author

Reviewer #1: This study reported the emergence of Omicron VOC and its rapid spread and persistence in the Western Amazon. It is a nice study with the clear methodology. There are a few comments:

1. What are subtypes of Omicron VOC?

2. What CoVID-19 vaccines did the patients receive?

3. How was the Illumina data analyzed to get complete genomes of SARS-CoV-2?

4. How was the phylogeny generated?

Reviewer #2: Summary:

This manuscript describes the collection and basic analyses of the emergence and spread of SARS-CoV-2 variants in Rondônia, Brazil. This piece highlights the increase in SARS-CoV-2 sequencing over the last months in Rondônia, Brazil and presents substantial new country-specific genomic data which is important in understanding the evolutionary mechanisms, transmission, and infectivity of SARS-CoV-2 variants in Brazil.

Suggestions:

1. It’s quite unfortunate that phylogeographic analysis was not included in this study which would have been used to infer the geographic origin of the variants, the course of range expansions, and the presence of genetic bottlenecks. Can this aspect be incorporated into the study?

2. Fig 3 shows three (3) clusters of BA.1 in the tree. Any explanation for the topology incongruence?

3. Your references are not consistent. Kindly check the PLOS format.

6. PLOS authors have the option to publish the peer review history of their article (what does this mean?). If published, this will include your full peer review and any attached files.

Reviewer #1: No

Reviewer #2: No

---

## [Author Response · Author response to Decision Letter 0]

25 Apr 2023

R: Manuscript was formatted in journal style.

Changes made in line 69 to 74: “All experiments were conducted in accordance with the relevant guidelines and regulations and were exempted from the requirements for informed consent due to the pandemic timing making direct contact with the subjects impossible. In addition, the study poses no risk to subjects due to the use of RNA from naso- or oropharyngeal samples already collected and extracted for the diagnosis of COVID by the centers, with no need for additional collections”.

 "This study was funded by Fundação Oswaldo Cruz de Rondônia – FIOCRUZ/RO, Departamento de Ciência e Tecnologia (DECIT), Fundação para o Desenvolvimento da Ação Científica e Tecnológica e à Pesquisa do Estado de Rondônia - FAPERO (Process:350.095.442.048.526.000.000/ 2016; Public bid invitation: 012/2016 PRO-RONDÔNIA and 001/2020 PPSUS) and by Instituto Nacional de Epidemiologia da Amazônia Ocidental - INCT EpiAmO. FGN is a CNPq fellow. Departamento de Ciência e Tecnologia (DECIT) of the Brazilian MoH, US/CDC and OPAS, Brazilian office."

R: Changes made in line 315 to 317: The following sentence was added “ The funders had no role in study design, data collection and analysis, decision to publish, or preparation of the manuscript.”

 "The present study was developed by a group of researchers from Laboratório de Virologia Molecular da Fundação Oswaldo Cruz, in Rondônia, with financial support from the Genomic Coronavirus Fiocruz Network, Departamento de Ciencia e Tecnologia (DECIT), Fundação para o Desenvolvimento das Ações Científicas 

e Tecnológicas da Pesquisa do Estado de Rondônia – FAPERO, Programa de Pesquisa para o SUS (PPSUS), as well as Instituto Nacional de Ciência e Tecnologia de 

Epidemiologia da Amazônia Ocidental – INCT- EpiAmo who have been important contributors to scientific development in the Amazon region. Collaboration from 

Coordenação de Aperfeiçoamento Pessoal de Nível Superior – CAPES, from whom some authors received financial aid (scholarships) during the production of this study, the Vice president of Vigilância em Saúde e Laboratórios de Referências of Fiocruz, Instituto de Biologia Molecular do Paraná (IBMP) and Laboratório Central de Saúde Pública de Rondônia (LACEN/RO) were essential for the development of the study"

 "This study was funded by Fundação Oswaldo Cruz de Rondônia – FIOCRUZ/RO, Departamento de Ciência e Tecnologia (DECIT), Fundação para o Desenvolvimento da Ação Científica e Tecnológica e à Pesquisa do Estado de Rondônia - FAPERO (Process:350.095.442.048.526.000.000/ 2016; Public bid invitation: 012/2016 PRO-RONDÔNIA and 001/2020 PPSUS) and by Instituto Nacional de Epidemiologia da Amazônia Ocidental - INCT EpiAmO. FGN is a CNPq fellow. Departamento de Ciência e Tecnologia (DECIT) of the Brazilian MoH, US/CDC and OPAS, Brazilian office."

R:The acknowledgements of the study's funding bodies have been removed, thus adjusting them to line 293 to 299: “The present study was developed by a group of researchers from Laboratório de Virologia Molecular da Fundação Oswaldo Cruz, in collaboration from Coordenação de Aperfeiçoamento Pessoal de Nível Superior – CAPES, from whom some authors received financial aid (scholarships) during the production of this study, the Vice president of Vigilância em Saúde e Laboratórios de Referências of Fiocruz, Instituto de Biologia Molecular do Paraná (IBMP) and Laboratório Central de Saúde Pública de Rondônia (LACEN/RO) were essential for the development of the study.” 

R: Changes made in line 308 to 317: “Funding: This study was funded by Fundação Oswaldo Cruz de Rondônia - FIOCRUZ/RO, the Genomic Coronavirus Fiocruz Network, Department of Science and Technology (DECIT), Foundation for the Development of Scientific and Technological Action and Research of the State of Rondônia - FAPERO (Process:350. 095.442.048.526.000.000/ 2016; Public bid invitation: 012/2016 PRO-RONDÔNIA and 001/2020 PPSUS) and by Instituto Nacional de Epidemiologia da Amazônia Ocidental - INCT EpiAmO. FGN is a CNPq fellow. Department of Science and Technology (DECIT) of the Brazilian MoH, US/CDC and PAHO, Brazilian office. The funders had no role in study design, data collection and analysis, decision to publish, or preparation of the manuscript.”

 "At the time of submission, R.C.P.R, L.G.M and F.K.M were employees at IBMP, which manufactures and commercializes the test described in this study. The other authors declare no potential conflict of interest."

R: Changes made in line 320 to 321: Inserting sentence at the end of the topic "This does not alter our adherence to PLOS ONE policies on sharing data and materials."

The information that will be passed on comprises the sequence IDs collected in GISAID, which is already in the manuscript as background material (S1 Table) line 390 to 391.

7. We note that you have included the phrase “data not shown” in your manuscript. Unfortunately, this does not meet our data sharing requirements. PLOS does not permit references to inaccessible data. We require that authors provide all relevant data within the paper, Supporting Information files, or in an acceptable, public repository. Please add a citation to support this phrase or upload the data that corresponds with these findings to a stable repository (such as Figshare or Dryad) and provide and URLs, DOIs, or accession numbers that may be used to access these data. Or, if the data are not a core part of the research being presented in your study, we ask that you remove the phrase that refers to these data.

R: "Data not shown" referred to the map (Figure 1). With the removal of the map the article has all data according to PLOS policy.”

8. Your ethics statement should only appear in the Methods section of your manuscript. If your ethics statement is written in any section besides the Methods, please delete it from any other section. 

R: Topic removed: “Institutional Review Board Statement: The project was evaluated and approved by the Research Ethics Committee of the Research Center for Tropical Medicine - CEPEM - Rondônia under protocol no. 4,000,086 and carried out in accordance with the ethical principles stipulated by the 1975 World Medical Assembly and the Ministry of Health (Resolution 466).”

9. We note that Figure 1 in your submission contain [map/satellite] images which may be copyrighted. All PLOS content is published under the Creative Commons Attribution License (CC BY 4.0), which means that the manuscript, images, and Supporting Information files will be freely available online, and any third party is permitted to access, download, copy, distribute, and use these materials in any way, even commercially, with proper attribution. For these reasons, we cannot publish previously copyrighted maps or satellite images created using proprietary data, such as Google software (Google Maps, Street View, and Earth). For more information, see our copyright guidelines: http://journals.plos.org/plosone/s/licenses-and-copyright.

“It was not possible to obtain the Creative Commons Attribution License (CC BY 4.0), so we have chosen to remove the map at this point (figure 1).”

10. Please include captions for your Supporting Information files at the end of your manuscript, and update any in-text citations to match accordingly. Please see our Supporting Information guidelines for more information: http://journals.plos.org/plosone/s/supporting-information. 

R: Changes made in line 480 to 484: S1 Table. SARS-CoV-2 genomes. All genome sequences and associated metadata in this dataset are published in GISAID’s EpiCoV database. To view the contributors of each individual sequence with details such as accession number, Virus name, Collection date, Originating Lab and Submitting Lab and the list of Authors, visit 10.55876/gis8.230107pn

Two new references have been added to the methodology about the genome assembly, so that references 20 and 21 have been added to the following section in the methodology: “FASTQ reads were generated by the Illumina pipeline in BaseSpace. Consensus sequences were generated using DRAGEN COVID LINEAGE 3.5.1 to 3.5.3, according to the most updated version of this application in each sequencing run. Subsequently, the quality of the consensus files was analyzed using the Nextclade v1.5.2 tool [20], those with more than 1% "Ns" ambiguities had the FASTQ files imported into Geneious Prime 2021 for trimming and assembly using a custom workflow employing the BBDuk and BBMap tools (v38.84) and the NC_045512.2 RefSeq as a template with careful visual inspection. Using both approaches, we generated consensus sequences with average depth coverage greater than 800X, excluding duplicate reads. The genome-wide consensus sequences of SARS-CoV-2 were initially assigned to viral lineages using the Pango Lineage web application [21].”

Reviewers' comments:

Reviewer's Responses to Questions

Comments to the Author

Reviewer #1: This study reported the emergence of Omicron VOC and its rapid spread and persistence in the Western Amazon. It is a nice study with the clear methodology. There are a few comments:

1. What are subtypes of Omicron VOC?

Information found on the page 198-205: Twenty-seven Omicron VOC subtypes were found: “The analyses of the sequenced samples corresponded to 30.14% (167) BA.1; 18.05% (100) BA.1.1; 6.14% (34) BA.1.1.1; 0.18% (1) BA.1.1.14; 0.18% (1) BA.1.1.15; 0.18% (1) BA.1.1.18; 1.26% (7) BA.1.14; 12.82% (71) BA.1.14.1; 0.18% (1) BA.1.14.2; 0.90% (5) BA.1.15; 2.17% (12) BA.1.17.2; 0.18% (1) BA.1.20; 0.36% (2) BA.1.9; 2.35% (13) BA.2; 0.18% (1) BA.2.12.1; 0.18% (1) BA.2.36; 0.72% (4) BA.2.56; 0.18% (1) BA.2.81; 1.81% (10) BA.4; 0.18% (1) BA.4.1; 9.93% (55) BA.5.1; 0.18% (1) BA.5.1.15; 8.84% (49) BA.5.2.1; 0.36% (2) BA.5. 6; 0.54% (3) BE.9; 1.62% (9) BF.5 and 0.18% (1) BQ.1.1. To date, no cases of XBB.1.5 have been reported in the state of Rondônia.”

2. What CoVID-19 vaccines did the patients receive?

R: Changes made on pages 183 to 185: “The immunizers received by the study population were COVISHIELD, Sinovac, Comirnaty® | Pfizer Brazil, and Janssen-Cilag”.

3. How was the Illumina data analyzed to get complete genomes of SARS-CoV-2?

R: Changes made on pages 134 to 144: “FASTQ reads were generated by the Illumina pipeline in BaseSpace. Consensus sequences were generated using DRAGEN COVID LINEAGE 3.5.1 to 3.5.3, according to the most updated version of this application in each sequencing run. Subsequently, the quality of the consensus files was analyzed using the Nextclade v1.5.2 tool [20], those with more than 1% "Ns" ambiguities had the FASTQ files imported into Geneious Prime 2021 for trimming and assembly using a custom workflow employing the BBDuk and BBMap tools (v38.84) and the NC_045512.2 RefSeq as a template with careful visual inspection. Using both approaches, we generated consensus sequences with average depth coverage greater than 800X, excluding duplicate reads. The genome-wide consensus sequences of SARS-CoV-2 were initially assigned to viral lineages using the Pango Lineage web application [21].”

4. How was the phylogeny generated?

Initially, high quality reference sequences were retrieved from all the VOC Omicron lineages found in the study and then the dataset was aligned using MAFFT. The phylogeny was generated based on the maximum likelihood method. We used IQ-TREE 2 with the Ultrafast Bootstrap algorithm which is already frequently used for phylogenetic analysis applied to SARS-CoV-2.

Reviewer #2: Summary:

This manuscript describes the collection and basic analyses of the emergence and spread of SARS-CoV-2 variants in Rondônia, Brazil. This piece highlights the increase in SARS-CoV-2 sequencing over the last months in Rondônia, Brazil and presents substantial new country-specific genomic data which is important in understanding the evolutionary mechanisms, transmission, and infectivity of SARS-CoV-2 variants in Brazil.

Suggestions:

1. It’s quite unfortunate that phylogeographic analysis was not included in this study which would have been used to infer the geographic origin of the variants, the course of range expansions, and the presence of genetic bottlenecks. Can this aspect be incorporated into the study?

Unfortunately, to perform a robust analysis there are some limitations related to the insufficient number of high quality deposited sequences of particular lineages. 

In addition, a long period of time will be required to run the analyses considering a large data set, which would not be possible for the proposed response period.

2. Fig 3 shows three (3) clusters of BA.1 in the tree. Any explanation for the topology incongruence?

The justification for this topology is that the BA.1 lineage is the common ancestor of other BA.1* lineages and widely dispersed in different locations.

This can be checked in: https://nextstrain.org/ncov/gisaid/global/6m?tl=pango_lineage

Moreover, the clusters were strongly supported by a satisfactory bootstrap value (>99).

3. Your references are not consistent. Kindly check the PLOS format.

The format of the references was revised according to the journal model.

---

## [Decision Letter · Decision Letter 1]

2 May 2023

The emergence of Omicron VOC and its rapid spread and persistence in the Western Amazon

PONE-D-23-01948R1

Dear Dr. Vieira,

We’re pleased to inform you that your manuscript has been judged scientifically suitable for publication and will be formally accepted for publication once it meets all outstanding technical requirements.

Kind regards,

Babatunde Olanrewaju Motayo, Ph.D.

Academic Editor

PLOS ONE

Additional Editor Comments (optional):

Reviewers' comments:

Reviewer's Responses to Questions

**Comments to the Author**

1. If the authors have adequately addressed your comments raised in a previous round of review and you feel that this manuscript is now acceptable for publication, you may indicate that here to bypass the “Comments to the Author” section, enter your conflict of interest statement in the “Confidential to Editor” section, and submit your "Accept" recommendation.

Reviewer #1: All comments have been addressed

Reviewer #2: All comments have been addressed

2. Is the manuscript technically sound, and do the data support the conclusions?

Reviewer #1: Yes

Reviewer #2: Yes

3. Has the statistical analysis been performed appropriately and rigorously? 

Reviewer #1: N/A

Reviewer #2: Yes

4. Have the authors made all data underlying the findings in their manuscript fully available?

Reviewer #1: Yes

Reviewer #2: Yes

5. Is the manuscript presented in an intelligible fashion and written in standard English?

Reviewer #1: Yes

Reviewer #2: Yes

6. Review Comments to the Author

Reviewer #1: (No Response)

Reviewer #2: The authors have made significant improvement on the manuscript and equally addressed most of the concerns raised during the initial review. Their inability to include the phylogeographic analysis is quite unfortunate but understandable due to time factor.

7. PLOS authors have the option to publish the peer review history of their article (what does this mean?). If published, this will include your full peer review and any attached files.

Reviewer #1: No

Reviewer #2: No

---

## [Editor Report · Acceptance letter]

4 May 2023

PONE-D-23-01948R1 

The emergence of Omicron VOC and its rapid spread and persistence in the Western Amazon 

Dear Dr. Vieira:

I'm pleased to inform you that your manuscript has been deemed suitable for publication in PLOS ONE. Congratulations! Your manuscript is now with our production department. 

Kind regards, 

on behalf of

Dr Babatunde Olanrewaju Motayo 

Academic Editor

PLOS ONE